# End-to-End QA with Polymer Gel Dosimeter for Photon Beam Radiation Therapy

**DOI:** 10.3390/gels9030212

**Published:** 2023-03-10

**Authors:** Libing Zhu, Yi Du, Yahui Peng, Xincheng Xiang, Xiangang Wang

**Affiliations:** 1Institute of Nuclear and New Energy Technology, Tsinghua University, Beijing 100084, China; 2Key Laboratory of Carcinogenesis and Translational Research (Ministry of Education/Beijing), Department of Radiotherapy, Peking University Cancer Hospital & Institute, Beijing 100142, China; 3School of Electronic and Information Engineering, Beijing Jiaotong University, Beijing 100044, China

**Keywords:** end-to-end QA, polymer gel dosimeter, one delivery phantom, photon beam

## Abstract

With the complexity and high demands on quality assurance (QA) of photon beam radiation therapy, end-to-end (E2E) QA is necessary to validate the entire treatment workflow from pre-treatment imaging to beam delivery. A polymer gel dosimeter is a promising tool for three-dimensional (3D) dose distribution measurement. The purpose of this study is to design a fast “one delivery” polymethyl methacrylate (PMMA) phantom with a polymer gel dosimeter for the E2E QA test of the photon beam. The one delivery phantom is composed of ten calibration cuvettes for the calibration curve measurement, two 10 cm gel dosimeter inserts for the dose distribution measurement, and three 5.5 cm gel dosimeters for the square field measurement. The one delivery phantom holder is comparable in size and shape to that of a human thorax and abdomen. In addition, an anthropomorphic head phantom was employed to measure the patient-specific dose distribution of a VMAT plan. The E2E dosimetry was verified by undertaking the whole RT procedure (immobilization, CT simulation, treatment planning, phantom set-up, imaged-guided registration, and beam delivery). The calibration curve, field size, and patient-specific dose were measured with a polymer gel dosimeter. The positioning error can be mitigated with the one-delivery PMMA phantom holder. The delivered dose measured with a polymer gel dosimeter was compared with the planned dose. The gamma passing rate is 86.64% with the MAGAT-f gel dosimeter. The results ascertain the feasibility of the one delivery phantom with a polymer gel dosimeter for a photon beam in E2E QA. The QA time can be reduced with the designed one delivery phantom.

## 1. Introduction

With the development of radiation techniques, the high demand for quality assurance (QA) increases accordingly. Even though the treatment machines and treatment planning system are verified with routine quality assurance (QA) testing, it does not inherently guarantee that the entire treatment workflow and beam delivery are accurate [1]. Therefore, initiating routine end-to-end (E2E) QA is significant to validate the overall treatment process [1,2,3,4].

A polymer gel dosimeter is a 3D technique to measure the dose distribution with high spatial resolution. Compared with the one-dimensional ionization chamber and two-dimensional high-resolution radiochromic and radiographic film, polymer gel dosimeter possesses the unique potential to measure the three-dimensional (3D) dose distribution with high resolution [5] for clinical investigation. Polymer gel dosimeters function as chemical dosimeters which will polymerize after photon or proton irradiation. It principally consists of distilled water, gelatine, monomer, and oxygen scavenger. After irradiation, distilled water is resolved into free radicals, hydroxide radicals, and hydride radicals mainly. These radicals will react with the monomer of the polymer gel dosimeter, resulting in polymerization. The polymerization increases with the irradiated dose until saturation. The cross-linker facilitates a chain reaction, contributing to the formation of a cross-linked and more stable polymer. Employing magnetic resonance imaging (MRI) readout [6,7], the correlation between polymerization and dose can be characterized. Spin-spin relaxation rate (R2) is utilized to represent polymerization. The correlation between R2 and dose is the so-called “calibration curve”. A polymer gel dosimeter is employed as a relative dose measurement and investigated widely in photon beam radiation therapy. After irradiation, the gel can be imaged with MRI [8], X-ray CT [9], optical CT [10] and ultrasound [11] to analyze the dose distribution. The first polymer gel dosimeter employed to measure the 3D dose distribution was composed of N,N’- methylene-bis-acrylamide (bis), acrylamide (AA), and agarose [12]. Recently, a co-polymer of poly(ethylene oxide)-block-poly(propylene oxide)-blockpoly(ethylene oxide) (Pluronic F-127, PEO-PPO-PEO) [13] or AUQAJOINT(R) [14] was proposed as a gel matrix, replacing the gelatine for 3D polymer gel dosimeter. A mixture of gelatine and agar was investigated with an arbitrary shape [15]. Polymer gel dosimeters with reduced toxicity were proposed [16,17]. Moreover, great interest is focused on the new monomer investigation, such as N-Vinylcaprolactam [18], N-(Isobutoxymethyl) acrylamide [19], and N-(3-methoxypropyl) acrylamide [20].

To compare the actual delivered dose with the planned dose, the end-to-end (E2E) dosimetry was verified by undertaking the whole RT procedure with a polymer gel dosimeter [21,22,23]. E2E QA was investigated for intensity-modulated radiotherapy with a PAGAT gel dosimeter [24]. A normoxic N-Vinylpyrrolidone polymer gel dosimeter was employed for error detection in the E2E test for CT-based brachytherapy [25]. Gossman [26] employed a tetrazolium salt-based gel dosimeter as an E2E dosimeter for stereotactic radiosurgery. However, the feasibility of MAGAT-f gel dosimeter for E2E QA for both intensity-modulated radiotherapy (IMRT) and volumetric-modulated arc therapy (VMAT) has not been investigated yet. In this research, a fast and convenient polymethyl methacrylate (PMMA) phantom holder was designed with the combination of polymer gel dosimeter for E2E QA.

The positioning error of the calibration phantom will contribute to the discrepancy of the measured absorbed dose of the calibration phantom. Consequently, the calibration curve is inaccurate with a systematic dose error. Any variation in the position of the phantom will cause a dosimetric and geometrical error in the dose distribution [27]. Special registration devices such as a stereotactic head frame [28,29] or an optical tracking system [30] can be employed to reduce the positioning error. When investigating the reproducibility of a polymer gel dosimeter, it is significant to keep the same gel setup position during irradiation. Deene et al. [24] investigated the inter-batch and intra-batch accuracy and precision of polymer gel dosimeters. The spherical flask filled with a gel dosimeter was fixated with a phantom holder during irradiation and MRI scanning. Thus, the necessary phantom should be employed for the gel dosimeter.

The novelty of the research is to design a fast “one delivery” PMMA phantom for the photon beam E2E QA test. The phantom is comparable in size and shape to that of a human thorax and abdomen. It is composed of ten small gel dosimeter inserts for measuring the calibration curve, two gel dosimeter inserts with a diameter of 10 cm for measuring the patient-specific dose distribution, and three gel dosimeter inserts for measuring the uniform square field. By making the radiotherapy (RT) plans, the dose can be delivered once. The cross curves are on the surface of the phantom which allows for the alignment with the laser to mitigate the positioning errors of gel dosimeters. The phantom was employed for the machine QA and patient-specific QA for the photon beam meanwhile. The calibration curve, field size, percent depth dose, and patient dose distribution were measured with a polymer gel dosimeter.

## 2. Results

### 2.1. Calibration Curve Measurement

The dose-response curve of MAGAT-f is shown in Figure 1. The irradiation dose (D) is 0, 2, 4, 6, 8, 10, 12, 14, 16, and 18 Gy. The cuvette without any irradiation acts as a control cuvette for measuring the background R2 value. The calibration phantom is scanned together with the large cylindrical phantom 24 h after irradiation. The standard deviation ranges from 0.0021 to 0.0125 s^−1^. The biggest standard deviation occurs at 12 Gy. The standard deviation can be attributed to the imaging noise. With the calibration curve, the R2 maps can be converted to dose maps to compare the TPS calculation.

### 2.2. Uniform Square Field

The 4 × 4 cm^2^ square field is shown in Figure 2 The dose distribution along the two profiles is in Figure 2b. After 3D volume rendering the field, a field size distribution can be obtained. The mean field size measured with MAGAT-f is 44.32 mm with a standard deviation of 2.70 mm. The deviation between the measurement and the theoretical value is 10.8%. The CNR is calculated based on the R2 map by choosing two regions of interest (ROI) as 58.62. The dashed square box is the two ROIs. The flatness and symmetry are 1.52% and 0.82%, respectively. Figure 3a illustrates the depth dose distribution of the square field. The percent depth dose (PDD) of both the sagittal plane and coronal plane is obtained in Figure 3b. The square field cross-section is tilted on the top region because of the oxygen contamination around the bottleneck. The sealing film was wrapped around the bottleneck. Obvious deviation occurs at the build-up region. The entrance region of the gel dosimeter is insensitive due to oxygen contamination. The temperature variation during irradiation and MRI scanning could affect the dose-response.

### 2.3. E2E QA with Polymer Gel Dosimeter

#### 2.3.1. IMRT Plan Verification

Figure 4 demonstrates the dose distribution of TPS calculation and gel measurement for the IMRT plan with five beams. TPS calculated dose distribution was based on the whole PMMA phantom so that the distal dose distribution can be seen (Figure 4a) while the gel dosimeter measured the dose distribution within the container with a diameter of 10 cm (Figure 4b). Figure 4c,d demonstrate the isolines of dose distribution of TPS and gel dosimeter measurement. Both doses were normalized to the tumor region so that the tumor region (red isolines) was equal to 1.

The images need to be registered because the irradiation setup is not the same as the MRI imaging setup. This is due to the gel being stored in the refrigerator at 0 °C after irradiation. Figure 5 demonstrates the procedure of image registration. The TPS images were calculated based on the whole PMMA phantom while the gel measurement is the plastic container area. Both TPS images and gel measurements were first cropped and resized to 512 × 512 images in order to investigate the dose discrepancy. The effective measuring diameter of the gel dosimeter was 10 cm. For the TPS images, an area with a diameter of 10 cm was cropped with pixels 39 × 39. Both TPS images and gel measurement images were normalized to 512 × 512 for comparison. The images of gel measurement were rotated to register with the TPS images. Finally, 2D gamma analysis was conducted to compare the delivered dose with the planned dose.

Figure 6 is the gamma analysis between TPS calculation and gel measurement with acceptance criteria 5%/3 mm. The gamma passing rate is 86.64%. For the gamma-index of more than 1, the point is considered “failed”. The main failing points occur in the targeted region. The dose deviation at the target region can be attributed to the dose-quenching effect of the polymer gel dosimeter. In addition, oxygen contamination in the gel dosimeter would also affect the dose-response even though the gel dosimeter was sealed with sealing film.

#### 2.3.2. VMAT Plan Investigation with an Anthropomorphic Head Phantom

The dose distribution of the brain tumor plan (VMAT plan) was measured with MAGAT-f. Figure 7a,b illustrate the dose map of MAGAT-f gel measurement and TPS calculation. The gamma analysis is employed to evaluate the discrepancy between measurement and calculation. The global gamma passing rate is 91.72% with criterion 3%/3 mm. The deviation between the measured dose and TPS calculation is attributed to two aspects. One is the discrepancy of the machine actual delivery parameters. The multi-leaf collimator position error will contribute to the delivery dose deviation. Another is the gel dosimeter stability. Several factors can affect the gel dosimeter response: preserving temperature, scanning temperature, and oxygen contamination. The preserving temperature is 4 °C in the refrigerator. During irradiation, the room temperature may affect the gel dosimeter. In addition, even though the gel dosimeter is normoxic, it does not mean the gel dosimeter will not be affected by oxygen. The oxygen contamination from the bottleneck will affect the dose-response.

## 3. Discussion

In this research, we investigated a fast end-to-end phantom with a polymer gel dosimeter by undertaking the whole RT procedure. The dose response is obtained by contouring 10 target regions based on the CT slices. Deene et al. [24] set up the test tube in the water tank for the irradiation. The uncertainty of the polymer gel dosimeter should be analyzed. The reproducibility will contribute to the measurement error. The reproductivity of a polymer gel dosimeter can cause errors in the measurement. Vandecasteele et al. [24] investigated the reproducibility study of a 3D polymer gel dosimeter (polyacrylamide gelatine gel). A high dosimetric precision (3.1%) is found for the intra-batch study. For the inter-batch experiment, the precision is 4.3%. Deene et al. [31] analyzed the detailed uncertainty of MAGAT and PAGAT gels. The overall accuracy was well within 5% when a careful scanning set-up was designed. In the experiments, the percent of gel and the storage and scanning temperature are carefully controlled. The reproducibility is under investigation by fabricating the gel in different batches to obtain the dose response after photon irradiation. The square field cross-section is tilted on the top region because of the oxygen contamination around the bottleneck. The sealing film was wrapped around the bottleneck. Obvious deviation occurs at the build-up region. The entrance region of the gel dosimeter is insensitive due to oxygen contamination. The temperature variation during irradiation and MRI scanning could affect the dose response.

The measured dose by polymer gel dosiemter was compared with the TPS calculation. The gamma passing rate is 86.64% with an acceptance criterion of 5%/3 mm for the IMRT plan while there are 91.72% points passing the acceptance criterion of 3%/3 mm for the VMAT plan. The deviation between the measured dose and TPS calculation is attributed to two aspects; one is the discrepancy of the machine’s actual delivery parameters. The multi-leaf collimator position error will contribute to the delivery dose deviation. The other is the gel dosimeter stability. Several factors can affect the gel dosimeter response: preserving temperature, scanning temperature, and oxygen contamination. The preserving temperature is 4 °C in the refrigerator. During irradiation, the room temperature may affect the gel dosimeter. In addition, even though the gel dosimeter is normoxic, it does not mean the gel dosimeter will not be affected by oxygen. The oxygen contamination from the bottleneck will affect the dose response.

Numerous pieces of research were conducted regarding the machine QA or patient-specific QA of photon beam radiation therapy with polymer gel dosimeters. The purpose of this study is to design a fast “one delivery and one scan” PMMA phantom. “One delivery” means that the ten calibration cuvettes were irradiated with ten radiotherapies (RT) plans without entering the treatment room to reduce irradiation time. In addition, the IMRT plan and the square fields were also delivered with two RT plans to maximally mitigate QA time. All the gel dosimeters were scanned with MRI once to obtain the R2 maps. Furthermore, a head phantom was employed for the E2E QA workflow verification of photon beam radiation therapy with a MAGAT-f gel dosimeter.

The size of the one delivery phantom can be enlarged to measure more different sizes of square fields meanwhile. The dose precision can be further improved with more strict gel dosimeter storage and shipping.

## 4. Conclusions

A fast convenient one delivery PMMA phantom was designed with a polymer gel dosimeter for photon beam radiation therapy. The purpose of the one delivery phantom is to irradiate the gel dosimeter with different RT plans. The end-to-end dosimetry was verified with a MAGAT-f gel dosimeter by undertaking the whole RT procedure (phantom set-up, CT simulation, contouring, treatment planning, image-guided registration, beam delivery, and MRI readout). The calibration curve, field size, percent depth dose curve, and patient-specific dose distribution were measured with a MAGAT-f gel dosimeter using different RT plans. The phantom can significantly reduce QA time because of the combination of the designed one delivery PMMA phantom and different RT plans. The R2 maps of the gel dosimeter can be obtained with MRI 3D readout in one scan to reduce the imaging time. The delivered dose measured with a polymer gel dosimeter was compared with the planned dose. The positioning error can be mitigated greatly with the dedicated PMMA phantom holder. Furthermore, the VMAT plan was verified by the head phantom with a gel dosimeter. The gamma passing rate is 91.72% (acceptance criterion 3%/3 mm) with MAGAT-f gel dosimeter for the VMAT plan. The gel dosimeter can be a promising tool for 3D dose verification and E2E QA workflow testing. Both QA time and MRI imaging time can be significantly reduced.

## 5. Material and Methods

### 5.1. E2E QA Procedure with Polymer Gel Dosimeter

Figure 8 illustrates the designed one delivery PMMA phantom with a polymer gel dosimeter for photon beam radiation therapy. The one delivery PMMA is composed of small gel dosimeters for measuring the calibration curve, three gel dosimeter inserts with a diameter of 5.5 cm for measuring the uniform square field and percent depth dose (PDD) curve, and two gel dosimeter inserts with a diameter of 10 cm for measuring the patient-specific dose distribution. The gel dosimeter was prepared first and inserted into the one delivery phantom (Figure 8a). The phantom was scanned with CT (Figure 8c) and the CT images were imported into the treatment planning system (TPS) for tumor contouring (Figure 8d). The phantom setup and beam delivery are in Figure 8e. After beam delivery, the PMMA phantom was scanned by (magnetic resonance imaging) MRI to obtain the R2 maps (Figure 8f).

Furthermore, E2E QA was investigated with a MAGAT-f gel dosimeter by undertaking the entire photon beam therapy process for a VMAT plan. The VMAT plan is measured with an anthropomorphic head phantom (Figure 9). The photon beam energy is 6 MV and the dose rate is 400 cGy min^−1^. After preparation, the gel dosimeter was scanned with CT. A brain tumor shape is contoured on the gel CT images with a target dose of 10 Gy (Figure 9c). The dose grid is set to 1.0 mm × 1.0 mm × 1.0 mm. The head phantom set-up is shown in Figure 9d,e. The phantom was scanned with MRI with a head coil to obtain the R2 maps (Figure 9f). With the calibration curve, the T2 map was converted to the dose map for the dose verification with the TPS calculation.

### 5.2. Gel Fabrication

The MAGAT-f gel dosimeter was employed in this work. The detailed formulation procedure can be found in [32]. The normoxic polymer gel dosimeter is composed of gelatine (8% *w*/*w*), distilled water (83% *w*/*w*), methacrylic acid (MAA, 6% *w*/*w*), formaldehyde solution (3% *w*/*w*), and 10 mM Tetrakis(hydroxymethyl)phosphonium Chloride (THPC). The mass concentration is shown in Table 1. The gelatine was first stirred and mixed in the distilled water by heating up to 48 °C with a water bath. After the gelatine was completely dissolved (taking 20 min), the gelatine resolution was cooled to 30 °C (30 min). MAA and formaldehyde resolution were added to the gelatine water solution. After 10 min of stirring, the antioxidant THPC is added. The gel was poured into ten glass cuvettes and plastic cylindrical containers. The bottleneck was wrapped with sealing film. The gel dosimeter was stored in the refrigerator at 4 °C. Ten 5 mL glass cuvettes with gel dosimeters were employed to measure the calibration curve. The gel dosimeter was sealed in cylindrical plastic containers with a diameter of 10 cm and height of 12 cm for the uniform field and patient-specific dose measurement.

### 5.3. Treatment Planning

The PMMA phantom with a polymer gel dosimeter is scanned with CT (Siemens, Berlin, Germany). The scanning parameters were: energy: 120 kVp, slice thickness: 5 mm, X-ray tube current: 136 mA, convolution kernel: B20s, matrix size (MS): 512 × 512, and pixel size: 1.2695 × 1.2695 mm^2^. The CT images were imported into the treatment planning system, Eclipse for the target contouring and planning. For the dose response measurement, ten 2 cm × 4 cm targets were contoured with a dose ranging from 2 to 18 Gy at the cuvettes region. The IMRT plan was composed of five beams with gantry angles 0°, 45°, 90°, 135°, and 179°, energy 6MV, and dose rate 400 cGy min^−1^. A lung tumor shape was contoured on the gel CT images with a target dose of 10 Gy. The dose grid was set to 1.0 mm × 1.0 mm × 1.0 mm.

### 5.4. Beam Delivery

All irradiation was performed on a clinical Varian linear accelerator (Varian Medical Systems, Palo Alto, CA, USA). For the dose response measurement, ten 2 cm × 4 cm targets are contoured with a dose ranging from 2 to 18 Gy at the cuvette region. The IMRT plan is composed of five beams with gantry angles 0°, 45°, 90°, 135°, and 179°, energy 6 MV, and dose rate 400 cGy min^−1^. A lung tumor shape is contoured on the gel CT images with a target dose of 10 Gy. A dedicated PMMA phantom holder with cross curves on the surface was designed to place the cylindrical phantoms and cuvettes. The PMMA phantom is aligned to the laser according to the cross curves. A CBCT system is employed for position registration before irradiation. All phantoms were irradiated without any repositioning of the phantom holder to minimize the set-up errors of the phantom positioning.

### 5.5. MRI Readout

Spin-spin relaxation rate (R2) [33,34,35] is used to characterize the dose response of a polymer gel dosimeter. To investigate the relationship between R2 and the irradiated dose, ten cuvettes were irradiated with photon beams. The R2 map was acquired by a 3.0T MRI (Philips Achieva TX) with a body coil for the PMMA phantom. The same dedicated phantom holder used during irradiation is used. Multiple spin-echo sequences with spin-echo 16 were employed for imaging. The gel is preserved in the MRI room before imaging. The gel dosimeter was scanned 24 h after irradiation. The scanning parameters [36] are: maximum echo time (TE): 360 ms, repetition time (TR): 3000 ms, echo spacing: 22.5 ms, and the number of excitations: 1. The MRI scanning parameters are in Table 2.

The gamma-index is calculated by Equation (1) to evaluate the measured dose with a polymer gel dosimeter and the planned dose in TPS.
(1)$$\gamma =\sqrt{\frac{{\left(\vec{r}-{\vec{r}}_{2}\right)}^{2}}{DTA_{a}^{2}}+\frac{{\left(D-D_{r}\right)}^{2}}{DD_{a}}}$$
where $\vec{r}$ and ${\vec{r}}_{2}$ are the position vectors of a point of the gel dosimeter measurement and the nearest point with the same dose in the TPS dose distribution (nominal dose) respectively. *D* and *Dr* denote the measured dose and planned dose at the position $\vec{r}$. *DTA_a_* and *DD_a_* are the acceptance criteria in spatial position and dose. By calculating the gamma-index pixel by pixel, the gamma map is then obtained.

## Figures and Tables

**Figure 1 gels-09-00212-f001:**
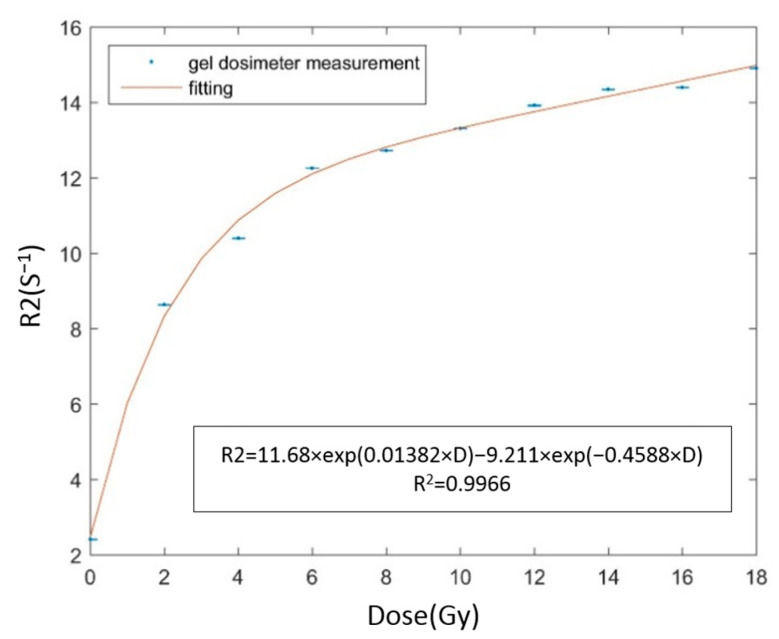
The dose-response curve of MAGAT-f gel dosimeter under photon beam irradiation.

**Figure 2 gels-09-00212-f002:**
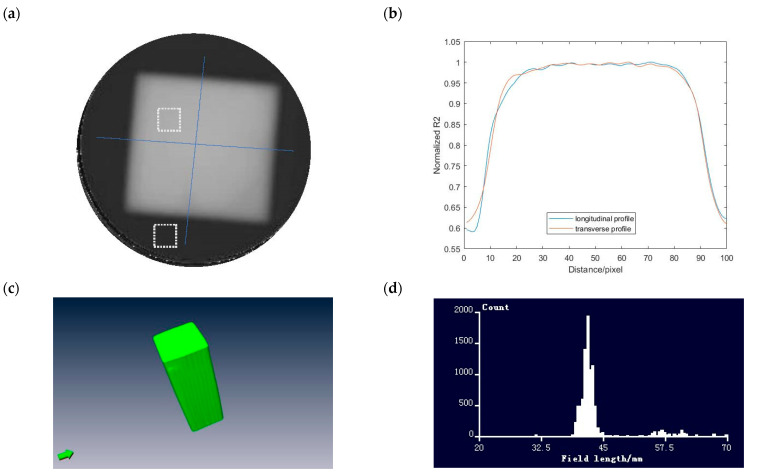
(**a**) R2 map of 4 × 4 cm^2^ uniform field measured with MAGAT-f (**b**) lateral R2 profile of the two blue lines (pixel = 0.5 mm), (**c**) rendered uniform field, and (**d**) field size distribution measured with MAGAT-f.

**Figure 3 gels-09-00212-f003:**
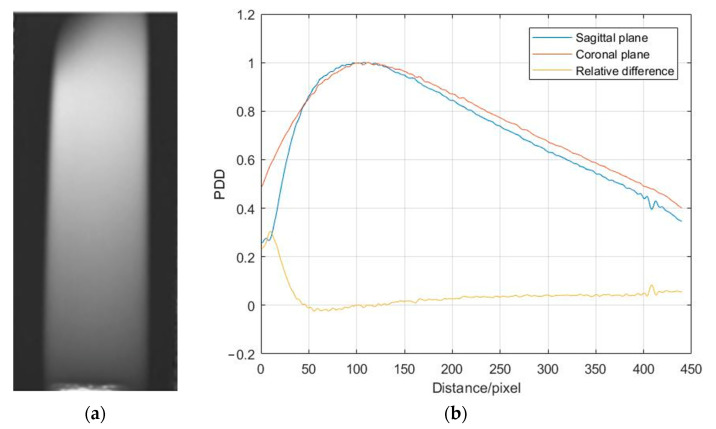
(**a**) R2 map of the depth dose distribution and (**b**) percent depth dose curve measured with MAGAT-f (pixel = 0.5 mm).

**Figure 4 gels-09-00212-f004:**
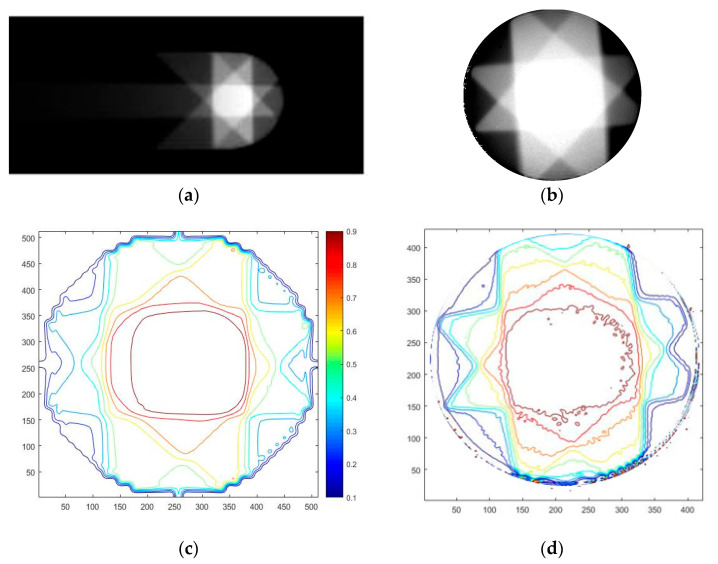
IMRT plan of (**a**) TPS calculation (**b**) gel measurement, isolines of (**c**) TPS calculation, and (**d**) gel measurement.

**Figure 5 gels-09-00212-f005:**
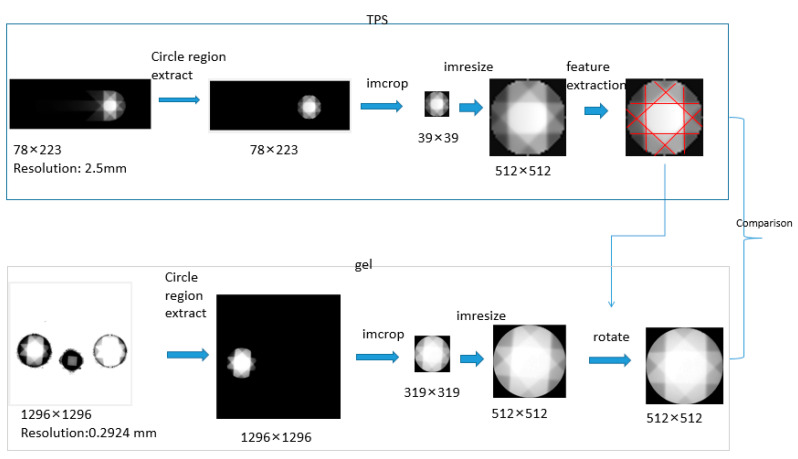
Image registration protocol for the polymer gel dosimeter measurement.

**Figure 6 gels-09-00212-f006:**
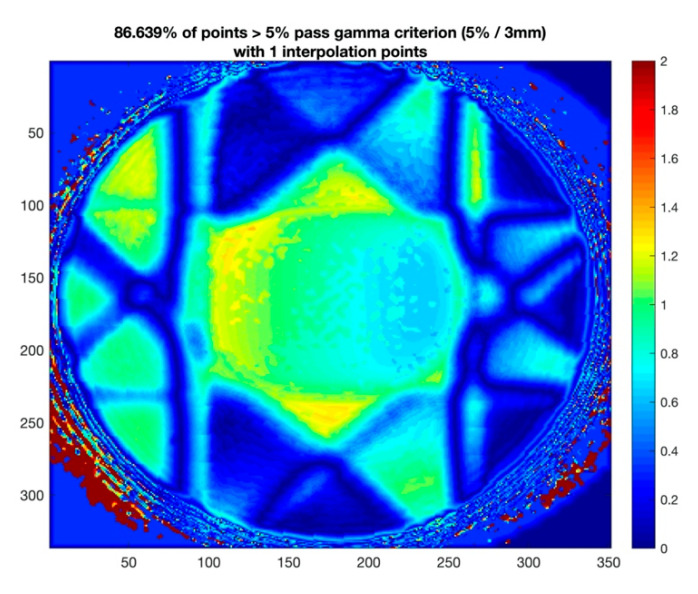
Gamma analysis between TPS calculation and gel measurement with criterion 5%/3 mm.

**Figure 7 gels-09-00212-f007:**
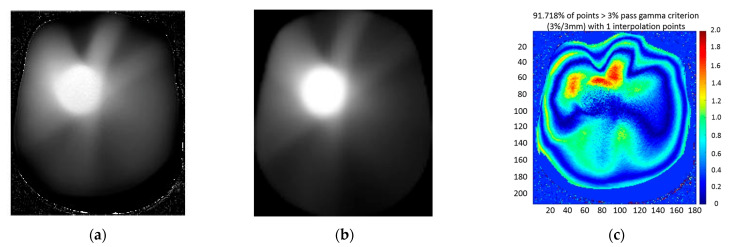
(**a**) MAGAT-f gel measurement, (**b**) TPS calculation of VMAT plan, and (**c**) 2D gamma map between gel measurement and TPS calculation.

**Figure 8 gels-09-00212-f008:**
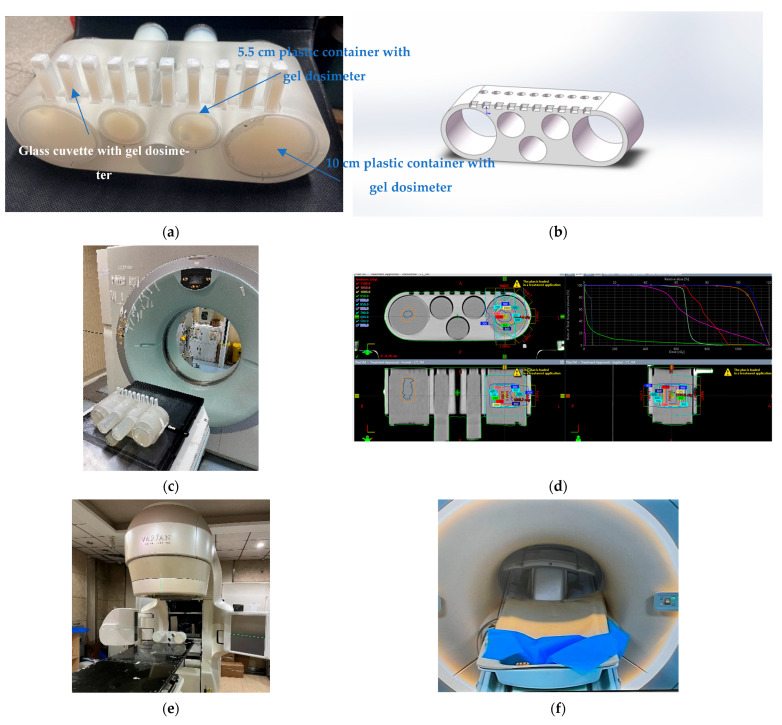
(**a**) A photographic representation of the PMMA phantom holder to fixate the cylindrical containers and cuvettes, (**b**) scheme of PMMA phantom holder, (**c**) CT simulation, (**d**) scheme of IMRT plan based on the polymer gel dosimeter, (**e**) irradiation setup, and (**f**) MR scanning with a body coil for the PMMA phantom holder with gel dosimeter.

**Figure 9 gels-09-00212-f009:**
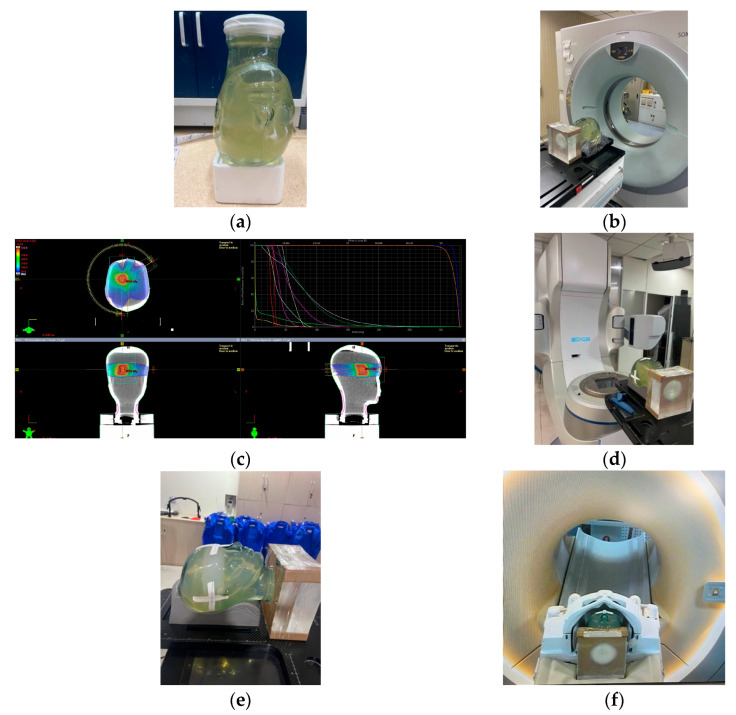
(**a**) A photographic representation of head phantom, (**b**) CT simulation, (**c**) VMAT plan of brain tumor based on the CT slices of polymer gel dosimeter, (**d**) irradiation setup, (**e**) MAGAT-f gel dosimeter after irradiation, and (**f**) MR scanning with a head coil for the head phantom.

**Table 1 gels-09-00212-t001:** Mass concentration of each polymer gel dosimeter.

Component	MAGAT-f
Distilled water	83%
Gelatin (Type A, 300 Bloom)	8%
Methacrylic acid 99.5%	6%
Formaldehyde solution with 37% minimum and stabilized with 10% methanol	3%
Tetrakis(hydroxymethyl) phosphonium Chloride (THPC) 80%	10 mM

**Table 2 gels-09-00212-t002:** MRI scanning parameters.

Parameter	Units	Value
Echo spacing	ms	22.5
Maximum echo time (TE)	ms	360
Repetition time (TR)	ms	3000
Field of view (FOV)	mm	180
Matrix size (MS)	pixels	512 × 512
Slice thickness	mm	3
Number of excitations (NEX)		1

## Data Availability

Data is contained within the article.

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
