# Peer review of "End-to-End QA with Polymer Gel Dosimeter for Photon Beam Radiation Therapy"

_gels, 2023, doi:10.3390/gels9030212_

Round 1
Reviewer 1 Report
Comments to authors are listed below:
· The introduction should extend with further information, which are related to the topic.
· The novelty and applications of this works should be reported clearly to the last paragraph of the introduction section.
· The discussion section is very short and further explanations and comparisons should be included.
· The conclusions require to be strengthened with some significant results reported in this paper.
Reviewer 2 Report
This paper reports on the “End-to-end QA with polymer gel dosimeter for photon beam radiation therapy”. The subject is interesting, however, it needs to be edited well.
I have few comments to the manuscript:
1. The introduction lacks a clearly defined purpose of the research. Please add.
2. In the methodology, add references to publications.
3. Combine the results with the discussion.
4. From the results, transfer the descriptive part of the methodology to Materials and Methods.
5. Expand the discussion.
6. Materials and methods are missing a description of the materials.
7. Edit the Materials and Methods section well.
Taking into account all comments the manuscript may be published in Gels after major revision.
Reviewer 3 Report
In this study, a fast convenient PMMA phantom was designed with a polymer gel dosimeter for photon beam radiation therapy. The RT procedure verified the end-to-end dosimetry with MAGAT-f gel dosimeter. The calibration curve, field size, and patient-specific dose distribution were measured with MAGAT-f gel dosimeter using different RT plans. It was found that the phantom can significantly reduce QA time due to the combination of the designed PMMA phantom and RT plans. The R2 maps of the gel dosimeter were obtained with MRI 3D readout in one scan. The delivered dose measured with a polymer gel dosimeter was compared with the planned radiation dose.
The manuscript is very well written, and organized. it contains new findings. All images are clear and required for the quality estimations of the presented measurement method.
The results are discussed regarding the application potential.
The conclusions include a summary of the findings.
Round 2
Reviewer 1 Report
No comments.
Reviewer 2 Report
Manuscript can by published in presened form in Gels.